# The Effect of Housing Environment on Commercial Brown Egg Layer Production, USDA Grade and USDA Size Distribution

**DOI:** 10.3390/ani13040694

**Published:** 2023-02-16

**Authors:** Benjamin N. Alig, Peter R. Ferket, Ramon D. Malheiros, Kenneth E. Anderson

**Affiliations:** Prestage Department of Poultry Science, College of Agriculture and Life Sciences, North Carolina State University, Raleigh, NC 27606, USA

**Keywords:** egg production, laying hens, brown egg layers, housing environments, management and production, cage-free, free-range, USDA egg quality

## Abstract

**Simple Summary:**

Over the past several years, consumer preference has shown a trend towards better perceived welfare environments for the animals away from price. The egg industry is adjusting to this shift. The aim of our study was to determine differences in production from brown egg layers in various housing environments to better understand how these hens respond. We found that commercial brown egg layers had optimum performance parameters in extensive environments such as free-range, whereas brown egg layers in intensive environments, such as barren colony cages, did not perform well. This information shows us that it is ill-advised to house brown egg layers in conventional cages or barren colony cages, as these hens performed better in environments with enrichments such as enriched colony cages or the free-range environment.

**Abstract:**

Consumer demand for retail cage-free eggs is driving the layer industry towards greater use of extensive housing environments. However, there is limited research on how these environments affect egg production characteristics of brown egg layers, as was the focus of this study. Five housing environments were evaluated under typical industry conditions, including conventional cages, enrichable colony cages, enriched colony cages, cage-free and free-range. Three different brown egg laying strains were housed in the different housing environments and managed according to standard husbandry practices and stocking densities. Data collection for the strains began at 17 weeks of age, with a base period of 28 days for feed weigh backs and egg quality assessments. Housing environment had a highly significant (*p* < 0.0001) effect on all egg production characteristics measured, including egg production rates (% hen-day and % hen-housed), feed consumption (g/bird/day), feed conversion (egg g/feed g), and mortality rate (%) as well as percent grade A, B, and loss. Previous research revealed better egg production metrics for white egg layers in caged environments than extensive environments. In contrast, we observed brown egg layers had optimum production results for the free-range housing environments, and the poorest performance in enrichable colony cages.

## 1. Introduction

Over the past several years, the poultry industry has changed production management practices to optimize production efficiency, economic sustainability, and animal welfare. The layer industry has gone through several major changes in housing systems over the past several decades, from the humble beginnings of the classic free-range system, the industry developed and improved hen housing system to today’s cage systems. Almost all commercial layer farms utilize some form of indoor housing, with most birds housed in cages and approximately 25% of layers in cage-free systems [1]. The egg industry has developed many housing design options for layer farms, but it is not always clear what housing system is best for the various production and market scenarios that may prevail at the time.

Recently, pressure from some consumer interest groups and subsequent governmental regulation constraints have caused the egg industry to conform in accordance with animal welfare demands and adopt alternatives to the conventional cage house systems. States such as California have banned the sale of eggs from standard cage systems within their state’s borders [2]. As an alternative to cages, farmers in California now must exclusively use cage-free and free-range environments, and other states now favor the use of colony cages and the cage-free system. Furthermore, it has been shown that consumers are becoming more interested in buying eggs produced in alternative housing systems, especially brown eggs, than they were in the past [3,4]. Moreover, many fast-food chains and large grocery retailers have committed to using only eggs from alternative housing systems in the future [5,6].

Regardless of the demand for these housing systems, the alternative housing systems are more expensive to manage than the conventional cage system commonly used in the egg industry today. As housing systems become less intensive and more extensive, they require more labor to operate and therefore, are more costly to operate [7]. However, these eggs from extensive housing systems can often be sold for higher prices due to consumer demand, which offset the higher production costs than eggs from conventional cage systems [8]. 

Although studies on alternative housing systems in the United States are severely limited, differences in production among brown egg layers in these housing systems were observed in research undertaken in the Middle East and the United Kingdom. The intensive systems apparently resulted in better productive performance than the extensive systems, but the extent of these differences is inconclusive [9,10,11,12]. However, other research seems to suggest that the free-range environments may increase production performance [13,14]. Unfortunately, no controlled replicated study has evaluated all the most common systems together. Furthermore, as genetic, nutrition and management technologies improve, production parameters across these systems change in response to these improvements [15]. Any variation between studies can lead to confounded results, so though similar research was done in the Middle East and the United Kingdom, layer housing system research should be done in different regions to avoid any confounding effects from genetics, nutrition or management practices between geographical locations [16]. Furthermore, Al-Ajeeli et al. [17] reported differences in diet composition, such as various diets between regions, which can cause a variance in the production of eggs. Finally, no other study that sought to determine differences in housing environments has evaluated USDA egg sizes and grades, which are important to marketing eggs in the United States market. We hypothesized that while brown egg layers will have superior performance in conventional cages, brown egg layers in the free-range environment will also demonstrate favorable egg production parameters. The objective of the study reported herein was to identify the significant effects of housing systems on the production characteristics and marketable egg quality of modern genetic strains of brown egg layers, that have been provided similar nutrition and husbandry practices. This research was designed to be a comprehensive study of brown egg layer performance in the most common housing systems used in the commercial egg industry, utilizing several common strains.

## 2. Materials and Methods

This trial was conducted at the North Carolina Department of Agriculture and Consumer Services, Piedmont Research Station Poultry Unit in Salisbury NC in conjunction with the 40th NC Layer Performance and Management Test [18]. All housing environments were located on the same property. Three brown egg laying strains, Lohman LB Lites (Lohmann Tierzuckt GmbH, Cuxhaven, Germany), Hy-Line Brown and Hy-Line Silver Brown (Hy-Line International, West Des Moines, IA, USA) were utilized for this trial. Because the objectives of this study were to evaluate the effect of housing systems on brown egg laying hens in general, strain effects are not identified in the data tables. The chicks were sexed, vaccinated, tagged with individual identification numbers, weighed and placed in pullet rearing houses in accordance with the laying environment in which they would be housed. To simulate traditional industry practices, the birds that were destined for caged, enriched or enrichable systems were reared in a pullet cage system, and those birds that were destined to be in the free-range or cage-free environments were reared in the same environment they would be in during the laying phase. During their 17th week of age, the pullets were moved from the pullet houses to the lay houses. Egg quality and feed weigh backs were done every 4 weeks during the laying cycle of 17 to 89 weeks of age.

Three different brown egg layer strains were selected for this study based on commercial availability. Table 1 shows the strains, and the number of replicates per environment. The hatched chicks were sexed according to breeder recommendations (feather, color or vent sexing) and the females were retained for the study. Each strain was identified by a strain code for record keeping and data analysis, which kept the strains anonymous during analysis. At 69 weeks of age, some birds were molted as part of another study. Because the effects of molting were not a concern of this study, the data from those molted birds was removed from analysis. The number of birds that remained in the study is designated by the numbers in parentheses in Table 1 for each housing system and strain. The rearing phase was performed in conjunction with the 40th North Carolina Layer Performance Test, which can be found in the grow report [19]. 

Hens were divided into 5 different housing environments: conventional cages (CC), enrichable barren colony cages (CS), enriched colony cages (ECS), cage-free (CF) and free-range FR. As discussed previously, Table 1 shows the number of replicates used for each environment. The CC laying system was designed as a standard closed-sided house with forced ventilation. The cages were arranged in 3 tiers and utilized manure belts under each tier. Each cage measured 40.6 cm (16 in) high by 50.8 cm (20 in) deep by 121.9 cm (48 in) wide, thus providing 6131.6 cm^2^ (6.66 ft^2^) for 12 hens per cage with a stocking density of 516 cm^2^ (80 in^2^) per bird. There were 12 replicate units of 2 cages of 12 hens (24 hens per replicate) used for this trial. Each replicate unit of birds was fed by a trough system located on the outside of the cage. Nipple drinkers in each cage provided the birds with water. 

A single windowless, force-ventilated house contained 18 replicate cages of each of the two types of colony cages. This house contained 3 tiered banks of either CS or ECS replicates. This house utilized a system that allowed for control of feed (amount and diet) to each replicate, and each cage was equipped with nipple waterers. There was no difference in size between CS and ECS replicates. Each cage was 53.3 cm (21 in) tall by 66 cm (26 in) deep by 243.8 cm (96 in) wide, thus providing 1.6 m^2^ (17.3 ft^2^) for 31 birds per cage at a stocking density of 516 cm^2^ (80 in^2^) of cage space per bird. The major differences between CS and ECS were the contents of the cages. CS cages were barren colony cages that contain only the feeder and waterer systems. In contrast, ECS replicates contained several environmental enrichments, including nest boxes, roosts and a scratch area. ECS replicates used in this study were like those used in the commercial layer industry. 

The environmentally controlled CF house was a high-rise design with slatted flooring and manure pit beneath. Each 2.43 m × 3.05 m (8 ft × 10 ft) pen had 7.4 m^2^ (80 ft^2^) of floor space, half of which were covered in shavings and half were slats. A total of 10 replicates of CF pens were used for this study. When the laying cycle began at 17 weeks of age, the replicate population was adjusted to 60 birds per pen with a flock density of 1233 cm^2^ (192 in^2^) per bird. After subtracting the space utilized for the feeder’s area per hen was 1141 cm^2^ (177 in^2^). The birds were provided with feeder and waterer space in accordance with UEP guidelines. Each hen was provided 16 cm of roosting space and each pen contained 12 nesting boxes lined with AstroTurf for a total of 1 nest box per 5 hens.

The FR houses were curtain-sided houses with slatted floors and an attached outdoor section (paddock), which was completely enclosed with wire and included a net covering the top. The slats allowed the waste to drop below the house for easy removal without disturbing the birds. Once the hens were fully feathered, the heaters turned on only when the temperature dropped below 7.2 C (45F) to keep the birds in their effective thermal neutral zone. Supplemental lighting was provided for the birds to match the lighting schedules of the other houses. At 12 weeks of age the pullets were allowed access to their paddock. Both the pullets and layers were fed ad libitum. Feed and water were provided for the birds inside the coop and in the paddock. Each replicate unit had 8 nipple drinkers inside the coop and 8 nipple drinkers outside in the paddock. Tube feeders were placed inside the coop, and a covered feeder was placed outside in the paddock. Both feeders combined allowed for 6.4 cm of feeder space per bird. Each hen was provided 16 cm of roosting space, and each pen contained 12 nesting boxes lined with AstroTurf for a total of 1 nest box per 5 hens. Each FR replica had identical dimensions. The 4 m × 2 m (12.1 ft × 6.6 ft) house portion of each replica measured 7.4 m^2^ (80 ft^2^) with 60 birds per pen, which yielded a house stocking density of 1141 cm^2^ (177 in^2^) per bird. Each paddock was measured at 18.3 m × 18.3 m (60 ft × 60 ft). To preserve forage quality, a diagonal fence split each paddock exactly in half, and every 4 weeks the hens were rotated into a different section. One week before the paddock rotation, the grass inside the unused pen was mowed to a height of 15 cm (6 in). The fences of the paddock were 1.8 m (6 ft) high with mesh covering the top to prevent the birds from escaping and predators from entering. This rotating paddock design allowed for a stocking density of 2.8 m^2^ (30 ft^2^) per bird in the paddock. 

The layers used in this study were fed a on a phase feeding program according to standard industry practices, consisting of several different dietary phases that met or exceeded NRC recommendations based on current production rates and feed consumption rates [20]. Table 2 shows the diets fed based on production and consumption, and Table 3 details the composition of these different diets. Hens in all the different housing system environments followed the same ad libitum feeding program. As a note, feed 1 was not used and therefore was not included in the nutritional analysis. All hens were provided the same supplemental lighting program throughout the duration of the trial. The lighting program followed standard industry recommendations. All hens were fed the same feed during the rearing and pullet phase [19].

Performance data was collected and analyzed by period, with each period lasting for 4 weeks (28 days), beginning at week 17 and ending at week 92. Data was analyzed utilizing JMP 15.2 using ANOVA, and treatments were determined statistically different from one another by using Tukeys HSD [21]. Housing system, age and the interaction of the two were the main effects. Strain effects were found to be significant; however, they were not found to impact the significance of treatment and therefore were not included in the model. As a note, in the results section below, treatment effects described as better, worse, higher or lower are assumed statistically significant (*p* ≤ 0.05). No outliers were excluded in this data analysis.

Hen housed production is the percentage of eggs produced based on the number of hens placed at the beginning of lay, according to the following formula: Hen Housed % = ((Total period eggs)/(Hens-Housed × 28)) × 100.

Hen-day production is the amount of eggs laid based on the number of hens still in the trial, after accounting for those removed because of morbidity or mortality. Hen-day production was calculated as follows: Hen Day % = ((eggs produced)/(period hen days)) × 100.

Feed consumption was calculated on an average of grams/hen/day basis. The feed allocated was weighed during each period, and a weigh back done at the end of each period. Feed consumption rate was calculated using the following formula: Feed Consumption = (total grams of feed consumed during the period)/(period hen days).

Feed efficiency was calculated as a function of the mass of eggs produced relative to the feed consumed by the experimental unit, as follows: Feed Efficiency = (Period egg production × average egg weight grams)/(period feed consumption).

Overall mortality rate was calculated as the percentage of mortality that accumulated per replicate unit over the 19 observation periods relative to the total hens housed, as follows: Mortality = ((Hens housed-hens remaining)/(hens housed)) × 100.

Mortality rate data was transformed by
Mortality transformation = ASIN(SQRT(mortality %/100)) + 1
in order to normalize the distribution prior to statistical analysis. Mortality included dead hens as well as hens that were culled due to injury or sickness. 

During the third week of each observational period, all the eggs produced by each treatment replicate unit within the previous 24-h were collected to determine average egg weight, and the proportion of USDA egg size and grade categories [22]. Sampled eggs were weighed as a composite, and then the average period egg weight was calculated.
Average egg weight = (total grams of eggs weighed)/(number of eggs weighed).

Weights of individual eggs from each of these replicate sample groups was determined and then segregated into the USDA egg size categories of Pee Wee (<42.6 g), Small (42.6–49.7 g), Medium (49.7–56.8 g), Large (56.8–63.9 g) and Extra Large (>63.9 g) [22]. Jumbo was not used for this study. The distribution of eggs within each USDA egg size category was calculated as a percentage of the total number of eggs collected within each treatment replicate group. These same sampled eggs were assessed by trained individuals for exterior and interior quality, based on the USDA egg grading manual [22]. The distribution of eggs within each USDA grade category was calculated as a percentage of the total number of eggs collected within each treatment replicate group. The percentages of grade B eggs and loss eggs were calculated in the same manner. 

Individual body weights were taken only once at the end of the study and measured in kg per bird. 25 hens per replicate were weighed for both the CF and FR environments, while 13 birds were weighed per replicate from the CC, EC, and ECS environments. 

## 3. Results

### 3.1. Hen Weight

Table 4 contains the average weights of hens at the end of the study. This study found a statistically significant difference (*p* = 0.0206) between housing environments with FR hens being heavier, at 2.21 kg/bird, than CC, CS and ECS hens at 2.07 kg, 2.05 kg and 2.06 kg per bird, respectively. Hens in the CF environment (2.1 kg per bird) were not significantly different than their counterparts in other environments. 

### 3.2. Hen Day Production

Overall hen-day production shown in Table 4, while production averages by age are illustrated in Figure 1. Housing environment had a significant effect (*p* < 0.0001) on hen-day production of brown egg layers. The CS, ECS, and FR hens had higher overall hen-day egg production than CF by 1.7% to 2.9%, and CC hens by 2.1% to 3.3%. Hen-day egg production rate by age generally followed a trend typical of the breeder standards, but some statistical differences were observed among the housing environments. CS, ECS and FR hens had the highest hen-day production for most of the study, although FR hens had lower hen-day production from weeks 17 until 24 than the other environments. CC also had the highest hen-day production until week 48, after which the CC hens had the lowest hen-day egg production. Finally, the CF hens had the lowest hen-day egg production between weeks 21 and 68, after which the CC hens had similar egg production to the FR hens. 

### 3.3. Hen-Housed Production

Average hen-housed egg production rate of hens housed in each environment over the entire cycle of lay is presented in Table 4, whereas hen-housed production rate by age is presented in Figure 2. Housing environment had a significant effect (*p* < 0.0001) on hen-housed production rate. Overall, FR hens had 7.1%, 8.1%, 3.4%, and 7.8% higher hen-housed production rate than hens in the CC, CS, ECS and CF environments, respectively. ECS hens also had higher hen-housed production rate than hens in the CC, CS and CF environments by 3.7%, 4.7%, and 4.4%, respectively. For brown egg layers, hen-housed production rate by age followed a typical trend; although, hen-housed production from CC and CS hens fell sharply after weeks 52 and 68, respectively. Finally, FR hens had lower hen-housed production until week 24. However, these hens had the highest hen housed production afterwards.

### 3.4. Feed Consumption

Housing environment had a significant effect (*p* < 0.0001) on feed consumption (Table 4). Overall, CS, ECS and FR hens consumed more feed than CC and CF hens by 6.4 g feed/bird/day to 7.7 g feed/bird/day. Illustration of feed consumption by age showed housing environment effects were apparent for all ages (excluding weeks 21–24 and 81–84) (Figure 3). Feed consumption was consistent over all ages (excluding the FR spike from weeks 25 until 40), until week 57 when consumption for all environments began to increase. CC and CF hens consistently consumed the least amount of feed through the whole trial. FR hens had the highest feed consumption from week 28 until 40, and then had similar consumption to ECS hens after. CS and ECS hens had similar feed consumption through most of the cycle; however, CS hens had higher feed consumption then the other environments from week 69 until week 80. 

### 3.5. Feed Efficiency

Average feed efficiency of hens in each housing environment throughout the lay cycle shown in Table 4, whereas feed consumption rate by age is illustrated in Figure 4. Housing environment had a highly significant effect (*p* < 0.0001) on feed efficiency. Overall, ECS hens had lower feed efficiency than CC, CF and FR hens by 0.013, 0.023, and 0.025, respectively. Furthermore, CC hens had lower feed efficiency than CF and FR hens by 0.013 and 0.015, respectively, while CS hens had lower feed efficiency than CF and FR hens by 0.022 and 0.024, respectively. Following the same trend as hen-day egg production, feed efficiency started low, peaked with egg production, and then decreased towards the end of the cycle. Feed efficiency by age remained consistent for CC, CS, ECS, and CF hens, although CC hens had higher efficiency from week 25 until 32. CC CS and ECS hens began to lose their efficiency after week 72 as well. Finally, FR hens had the highest efficiency after week 36, but began to lose efficiency after week 76.

### 3.6. Egg Weight

Housing environment had a significant effect on egg weight, as seen in Table 4. Egg weight over time is presented in Figure 5. Overall, FR hens laid the heaviest eggs (*p* < 0.0001), with an average weight of 63.9 g per egg, while hens kept in CC laid the lightest eggs at 60.5 g per egg. Eggs from hens in CS, ECS and CF were statistically lighter than eggs from the FR environment, but heavier than eggs from CC at 61.1 g, 61.1 g and 61.3 g, respectively. By age, starting at week 25 and ending at week 68, birds from the FR environment consistently laid heavier eggs than birds in all other environments. While hens in the other environments increased their egg weight as they aged, the egg weights of FR hens remained consistently high for most of the study. 

### 3.7. Mortality

The effect of housing environment on total mortality rate is presented in Table 4. Although mortality rate was not summarized and statistically analyzed by age as for the other production parameters, the hen-housed egg production data (Figure 2) indicated that much of the housing effect on mortality rate occurred after week 56. Housing environment was found to have a highly significant effect (*p* < 0.0001) on total mortality. The CS hens had the highest mortality (36.9%), matching their inferior hen-housed production rate.

### 3.8. USDA Egg Grades

USDA egg grades are presented in Table 5, and data by age is shown in Figure 6, Figure 7 and Figure 8. Overall, hens produced the most grade A eggs, followed by loss eggs and the least amount of grade B eggs. Housing environment had a significant effect (*p* < 0.0001) on grade A and loss egg percentage where CF and FR hens laid the most grade A eggs by 6.1% to 9.8% and the least loss eggs by 6.7% to 8.8%. CC hens also produced more grade A eggs than CS hens by 1.6% and ECS hens by 2.4%. However, CC hens only produced less loss eggs than ECS hens by 2%. CS hens were not different from ECS or CC hens in terms of loss eggs. Housing environment also had a significant effect (*p* < 0.05) on grade B production, where CF hens produced more grade B eggs than CC hens by 0.63% and FR hens by 0.73%.

By age, grade A production seemed to slightly decline as the hens aged for CC, CS and ECS while grade B and loss inclined, whereas CF and FR grade A and loss production remained consistent and grade B inclined slightly after the first half of the trial. CF and FR hens consistently had the most grade A and least loss eggs during every significant age range. CC had similar grade A and loss production to CS and ECS for some age ranges, but also a higher grade A and lower loss for other age ranges. Both colony cages performed similarly, with CS hens laying more grade A in the beginning and less at the end. There was only one significant age range for medium egg production where CS hens laid more grade B eggs than CC and ECS hens; however, all hens except for CC hens spiked during this age range.

### 3.9. Egg Size Distribution

USDA egg size distribution data is presented in Table 5, and data by age was presented in Figure 9, Figure 10 and Figure 11. Housing environment had a significant effect on XL production (*p* < 0.0001), L production (*p* < 0.0001) and M production (*p* < 0.001). Overall, hens laid more XL eggs than L eggs, more L eggs than M eggs, and more M eggs than S eggs. Production of PW eggs was negligible for all treatments. FR hens laid the most XL eggs by 12.8% to 18.3%, and the least L eggs by 11.9% to 16.7%. Conversely, CC hens laid the least XL eggs by 3.6% to 18.3%, and the most L eggs by 3.8% to 16.7%, while CS, ECS, and CF hens fell between both XL and L. FR hens also laid less M eggs than CC hens by 1.5%, and ECS hens by 1.8%. By age, the XL production trend was upward for most of the study while L trended downward. M egg production was highest in the beginning then trended to almost zero for the rest of the study. FR hens produced the most XL eggs and the least L eggs for the first three quarters of the study. CS, ECS, and CF hens produced a similar amount of XL and L eggs to each other for the whole study. CC hens had similar XL and L proportions to CS, ECS and CF hens for the beginning. However, CC hens laid more L eggs and less XL eggs than the other treatments for many weeks in the middle of the study.

## 4. Discussion

The egg industry is moving towards greater usage of extensive housing environments due to pressures from consumer groups, large retailers, and government regulators. Regardless of the housing environment, economic sustainability of the commercial egg industry is dependent upon their profit margins (income driven by production metrics) over production costs (driven mostly by feed, housing, and labor costs). Therefore, it is most important to perform comparative studies that evaluate the production metrics of intensive and extensive layer housing environments, since production costs can more easily be determined. However, past research has reported conflicting results on the effect of different housing environments on layer performance characteristics [10,23,24]. This current study evaluated five of the major production environments and found significant differences between them in all parameters.

### 4.1. Egg Production

In this present study, hens in both CS, ECS and FR had higher hen-day production than CC and CF hens. Furthermore, FR hens had the highest hen-housed egg production rate, followed by ECS hens, with the other environments resulting in inferior hen-housed egg production. Dikmen et al. [13] found that egg production was indeed higher in FR brown egg layers. However, our study disagrees with Dikmen, who found that ECS hens had similar hen-day egg production to FR hens. Furthermore, Ahammed et al. [10] reported that there was no significant difference between CC hens, CF hens, and hens from aviaries, which is directly contradictory to our study. Furthermore, our study directly disagrees with Golden et al. [12] who reported CC hens had higher egg production than FR hens, which was due to heavy mortality from predation that the FR hens suffered. Our study also agrees with Bailey et al. [25] and Englmaierová et al. [26] who found that CF hens also had low egg production. When comparing the hen-day production between the two colony cage environments in our study, enrichments apparently had no beneficial effect, as observed by Onbaşılar et al. [27]. The difference in results between our study and the others could be due to confounding factors among the different regions, such as differences in genetics, nutrition, and management.

Interestingly, even though both hen-housed and hen-day egg production of the FR hens was highest among the housing environments in our study, FR hens took longer to reach peak egg production than the other environments, but subsequently maintained greater persistency of egg production than hens in the other housing environments. The reason FR hens took longer to reach peak production could be due to nutrient allocation to activity rather than production, slight stress when the pullets were allowed outside or the difference in light intensity between a standard house and the outdoor environment. Dikmen et al. [13] found similar results with their FR hens, reporting they peaked later but held their peak for longer. Although the FR hens took longer to reach peak production, this delay could have given these hens a slight developmental advantage in reproductive capacity. Anderson and Adams [28] also found that a delay in maturity lead to larger eggs in laying hens. Furthermore, studies with broiler breeders have shown that delayed sexual maturity causes these hens to have a more developed reproductive tract, heavier eggs and lower egg production [29,30]. Broiler breeders, unlike table egg layers, are not fed ad libitum and therefore may not have the nutrient intake required to produce both heavier eggs and more eggs, simultaneously. In contrast to broiler breeders, layers are genetically selected to lay eggs rather than muscle growth, and they do not need to be feed restricted to manage body weight. Therefore, we hypothesize that the reason the FR hens had higher egg production than the other environments was due to the delayed onset of lay, which allowed the FR hens to stock more resources into body reserves and more time to develop their reproductive tract. It appears that the CF hens are consistently the poorest performers, not only in our study but also in others such as Englmaierová et al. [26] who found that CF hens were consistently poor layers across the cycle. Englmaierová attributes lower production in the CF environment partly to the potential for hens to eat eggs in this system, which could partly explain the difference in the current study. Interestingly, even though CF and FR are similar environments, the FR hens in our study had higher hen-day egg production than CF hens. The major difference between these environments is that FR hens had access to an outdoor range, while CF hens did not. The range not only gave the FR hens access to natural sunlight but also access to forage [31]. Studies have shown that differences in light intensity and wavelength can affect egg production [32]. Therefore, hen access to sunlight could positively affect egg production in brown egg layers. 

### 4.2. Feed Consumption

The present study found that CS hens, ECS hens, and FR hens consumed the most feed, whereas CC hens and CF hens consumed the least amount of feed. Hens in extensive environments have been shown to be more active and less efficient in converting gross energy into metabolizable energy, and therefore must consume more feed than hens in intensive environments to meet energy demands [33]. CF hens, which were under similar conditions to FR hens, did consume less feed; however, the CF hens produced less eggs than CS, ECS or FR hens. The difference in feed consumption and egg production between the different housing environments seen in our study could indicate that feed consumption is not only affected by environment, but also by egg mass output, which is a function of production rate and egg weight. The data from our study also agreed with Karcher et al. [34], who found that CC hens ate less feed than ECS hens, and van Krimpen et al. [33], who reported hens in more extensive environments consumed more feed [33,34]. Furthermore, our study also agrees with other studies showing CC hens consume less feed than other cage environments. In contrast, our study does show that feed consumption of CF hens was similar to CC hens, which may be due to the low egg production and thus lower nutritional demand from the CF hens. Our study observed that hens in both colony cage environments had similar feed consumption rates, which agrees with other studies that compared these types of environments [23,27].

### 4.3. Feed Efficiency

CF and FR hens in our study had the highest feed efficiency, followed by the CC, CS and ECS hens, respectively. As observed for egg production, FR hens started off with poor feed efficiency, commensurate with their low egg production rates, but then they maintained high feed efficiency for the rest of the laying cycle. Both CS and ECS hens had similar efficiency as the rest of the housing environments, until the last quarter of the study when they both declined significantly. FR hens were unique in that they had access to forage to supplement their feed. It is known that forage and high fiber nutrients not only improves gut motility and gut health, but also nutrient absorption and digestibility [31,35]. Therefore, we hypothesize that high fiber forage, which the FR birds had access to, aided feed efficiency by allowing them to utilize more nutrients that they consumed for egg production. Perhaps if hens from other environments were given access to forage like the FR hens, they would respond with better feed efficiency. 

The results from our study agree with Dikmen et al. [13] who found that FR hens had the highest feed efficiency. In contrast, Al-Ajeeli et al. [17] observed that CC hens had better feed efficiency than FR hens. These different observations among researchers may be due to differences in forage quality and access to forage, which were not described. Englmaierová et al. [26] partly agrees with the findings of our study in that CC hens were observed to have better feed efficiency than hens in either colony cages. CF hens in our study had desirable feed efficiency, which was unexpected and contrary to the observation of Englmaierová et al. [26]. This improved feed efficiency would indicate that the CF environment does not result in increased expenditures of energy, as previously understood. As observed by Dikmen et al. [13], our study demonstrated that CS hens and ECS hens had similar feed efficiency. In contrast, Onbaşılar et al. [27] observed ECS hens had better feed efficiency than CS hens, which could be due to different cage designs than the ones in the present study. From our results, we hypothesize that adding enrichments to colony cages do not change the energy demands of the hen; however, more research, such as hen behavior, is needed.

### 4.4. Mortality

The mortality data from our study showed a difference between FR hens and CS hens, with FR hens having substantially lower mortality than CS hens. This difference in mortality could be partially due to the microbiome benefits that FR hens receive such as competitive inhibition. Furthermore, this could also be explained by differences in behavior and between the environments. These results agree with several other studies that show no difference in mortality among housing environments, omitting the comparison between FR and ECS [10,17,34]. Dikmen et al. [13] also found a difference in mortality between ECS and FR hens, however our study showed a much greater difference in mortality between ECS and FR hens. This perhaps indicates that larger flock sizes without enrichments can be detrimental to hens, although more research is needed.

### 4.5. Egg Weights

The few research reports that assessed egg production of different layer housing systems have shown conflicting or variable results related to egg quality [9,10,36]. Housing system had a significant effect on the average weight of eggs produced by brown egg layers, which contrasts with several other studies reporting no difference in egg weight from hens housed in different systems [10,17,37]. In agreement with our study, Dikmen et al. [13] observed FR hens laid heavier eggs than hens in either colony cage system, which was attributed to heavier bird weight. Al-Ajeeli et al. [17] also found that CC hens produced eggs of lower average weight than FR hens. Our study did not find any difference in average egg weight produced by hens in CS cages and hens in ECS cages. This is consistent with other studies that observed no difference in eggs weights from hens in colony cages, with or without enrichments [13,23,27]. In our study, the FR hens reached peak production at a later age than the other environments. Robinson et al. [29] found that when broiler breeder hen egg production is delayed, these hens have a more developed and heavier reproductive tract, and Joseph et al. [30] found that a delay in egg production causes broiler breeders to lay heavier eggs. Therefore, we believe that FR hens laid larger eggs due to the delay in sexual maturity, and we hypothesize that FR hens had larger reproductive organs as well.

### 4.6. USDA Egg Grades

Another novel contribution of this study was the observed differences in the distribution of USDA egg grades as influenced by the housing system. FR and CF hens had the highest proportion of grade A eggs, followed by CC hens, with hens from either colony cage system having the lowest proportion of grade A eggs. Conversely, CC, CS and ECS hens had a higher proportion of loss eggs than CF and FR hens. One reason why FR and CF hens produced more grade A eggs may be attributed to the nesting areas specific to these environments, as FR and CF hens can lay eggs on softer material. Furthermore, FR and CF hens had access to their eggs before collection while then hens in CC, CS or ECS systems did not. Therefore, there is a possibility that these hens ate any broken or cracked eggs that they produced. This study did not identify the specific reasons for the generation of loss eggs, which would be worthy of future study. In agreement with our research, Ahammed et al. [10] observed CC hens laid more cracked and dirty eggs than those hens in aviary or FR environments. On the other hand, Dikmen et al. [13] observed FR hens laid more dirty eggs, but fewer damaged eggs than those in CS and ECS hens. Onbasılar et al. [27] also observed brown egg layers in both colony cage systems laid the same amount of cracked and dirty eggs. 

### 4.7. USDA Egg Sizes

Our study also identified differences in the distributions of USDA egg sizes among housing systems. The USDA egg size classifications are used to determine the sales price of table eggs for consumers. Larger sized table eggs generally command a higher sales price than smaller sized eggs [38]. In our study, CC hens laid proportionally fewer XL-sized eggs and more L-sized eggs than hens in the other housing system environments. In contrast, the FR hens laid proportionally more XL-sized eggs and less L-sized eggs than the hens in the other housing system environments. While the FR hens laid the lowest number of L eggs, the XL eggs produce more material for further processing. Furthermore, while consumer preference leans towards L eggs, consumers may not care as much about the size of the egg compared with the housing environment. Further research is required to substantiate these theories. Finally, this study reported no differences in egg size distribution among eggs from CS and ECS systems, possibly indicating that the addition of enrichments does not affect the egg size distribution. The reporting of USDA egg size distribution for different housing systems is a novel contribution to the layer industry, especially for brown layers.

## 5. Conclusions

We originally hypothesized that hens in the CC and FR environments would produce superior production characteristics. While we found that hens from the CC environment had poorer production metrics, we also observed that FR environment enjoyed the superior production parameters. This indicates that access to high fiber forage from the range, the natural sunlight from the range and enrichments in the house can benefit the overall production of brown egg layers. Moreover, results from this study also indicated that hens in the ECS environments enjoyed a more optimal production performance than hens in CS cages, indicating that by simply adding enrichments can positively benefit brown egg layers. Finally, this research discovered that CF and FR hens produced a higher percentage of grade A eggs compared to the other environments, which could be contributed to gentler nesting material. This potentially indicates the need for improved flooring technology in cages, as many eggs are lost possibly due to the hard wire floor. In conclusion, this study did find significant differences in brown egg layer production parameters between housing environments. Therefore, these differences should be considered when designing and constructing housing environments for commercial egg farms utilizing brown egg layers.

## Figures and Tables

**Figure 1 animals-13-00694-f001:**
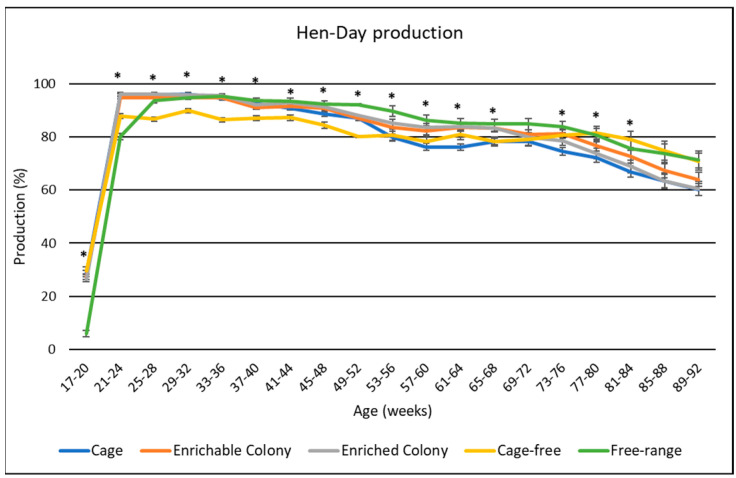
The effect of housing environment on hen–day production of brown egg layers by age. * Signifies a significant effect (*p* < 0.05) of housing environment during that age range.

**Figure 2 animals-13-00694-f002:**
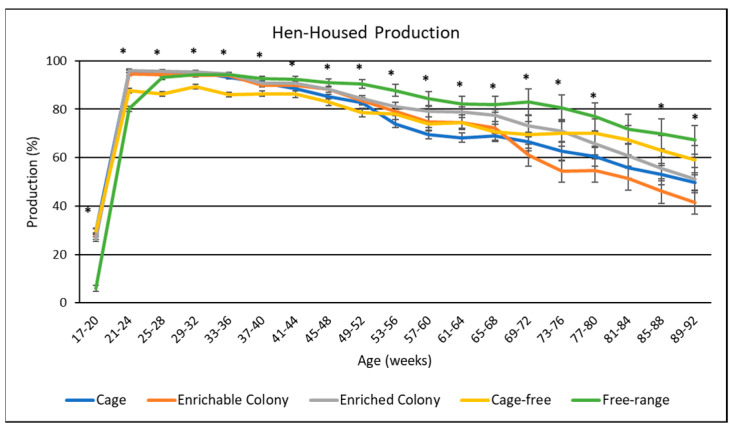
The effect of housing environment on hen–housed production of brown egg layers by age. * Signifies a significant effect (*p* < 0.05) of housing environment during that age range.

**Figure 3 animals-13-00694-f003:**
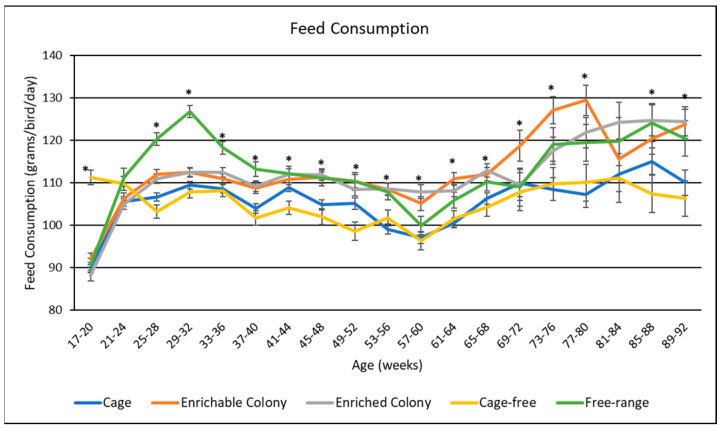
The effect of housing environment on feed consumption of brown egg layers by age. * Signifies a significant effect (*p* < 0.05) of housing environment by age range on feed consumption.

**Figure 4 animals-13-00694-f004:**
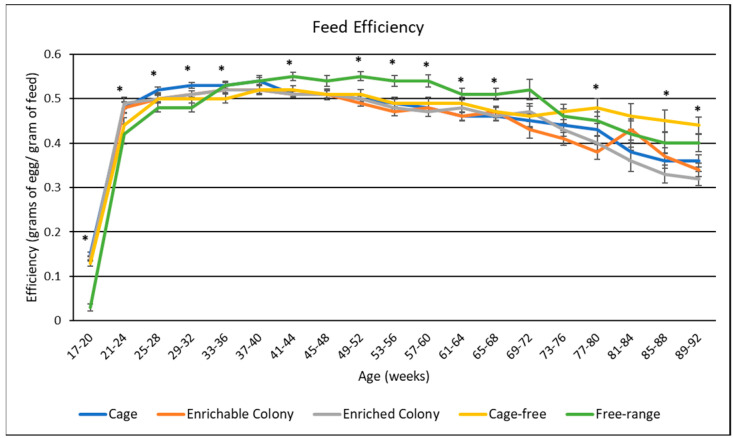
The effect of housing environment on feed efficiency of brown egg layers by age. * Signifies a significant effect (*p* < 0.05) of housing environment by age range on feed efficiency.

**Figure 5 animals-13-00694-f005:**
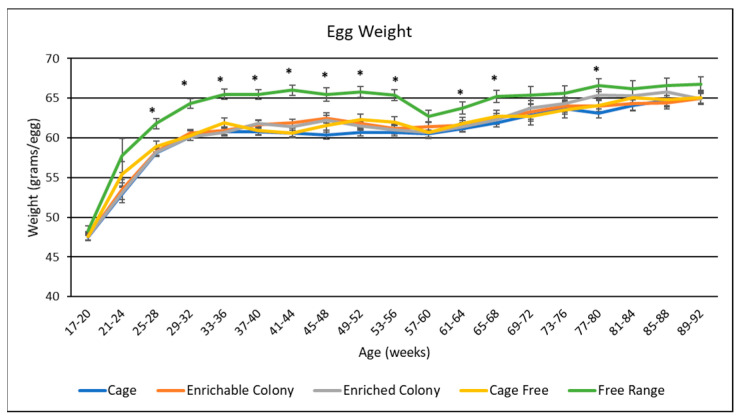
The effect of housing system on egg weight of brown egg layers by age. * Signifies a significant effect (*p* < 0.05) of housing environment by age range on egg weight.

**Figure 6 animals-13-00694-f006:**
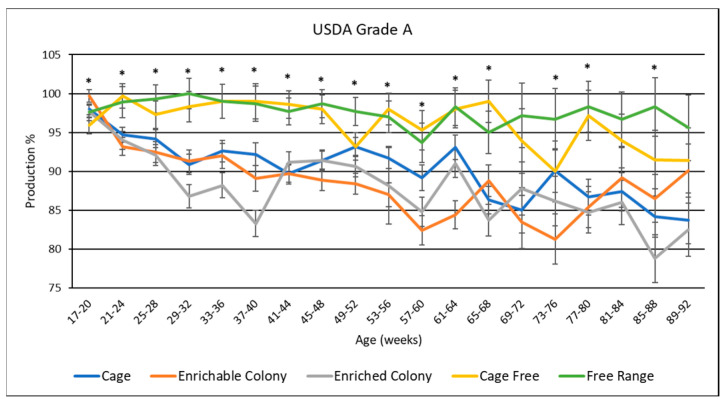
The effect of housing system on USDA grade A egg production of brown egg layers by age. * Signifies a significant effect (*p* < 0.05) of housing environment by age range on USDA grade A percentage.

**Figure 7 animals-13-00694-f007:**
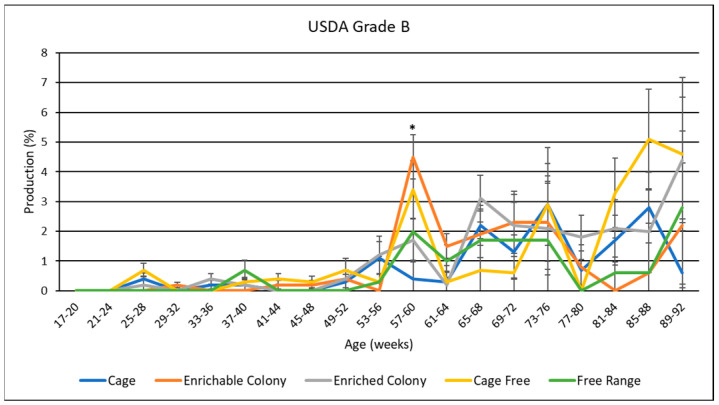
The effect of housing system on USDA grade B egg production of brown egg layers by age. * Signifies a significant effect (*p* < 0.05) of housing environment by age range on USDA grade B percentage.

**Figure 8 animals-13-00694-f008:**
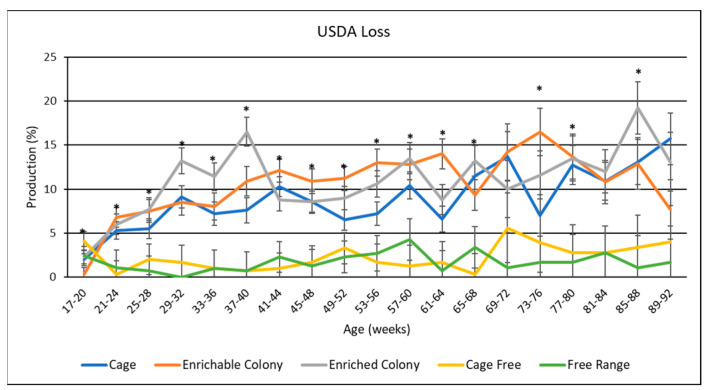
The effect of housing system on USDA grade loss production of brown egg layers by age. * Signifies a significant effect (*p* < 0.05) of housing environment during that age range on USDA grade loss percentage.

**Figure 9 animals-13-00694-f009:**
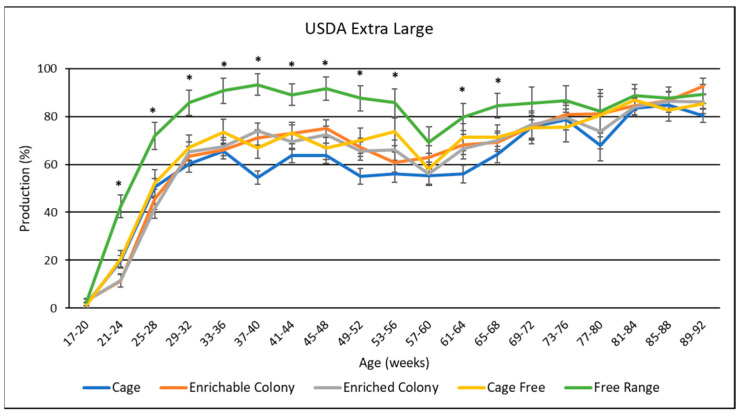
The effect of housing system on USDA extra large production of brown egg layers by age. * Signifies a significant effect (*p* < 0.05) of housing environment during the age range on USDA extra–large percentage.

**Figure 10 animals-13-00694-f010:**
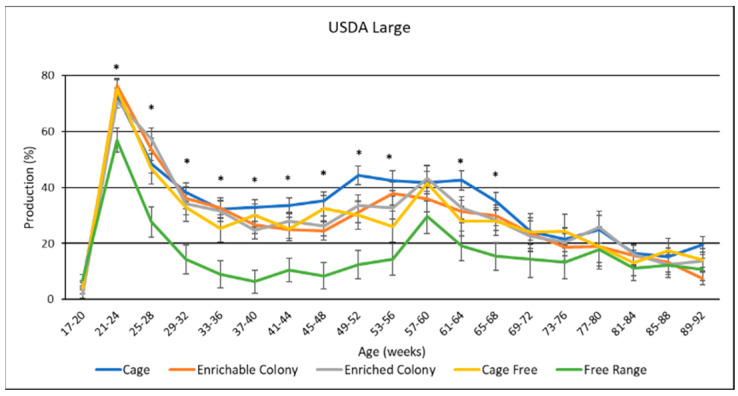
The effect of housing system on USDA large production of brown egg layers by age. * Signifies a significant effect (*p* < 0.05) of housing environment during that age range on USDA large percentage.

**Figure 11 animals-13-00694-f011:**
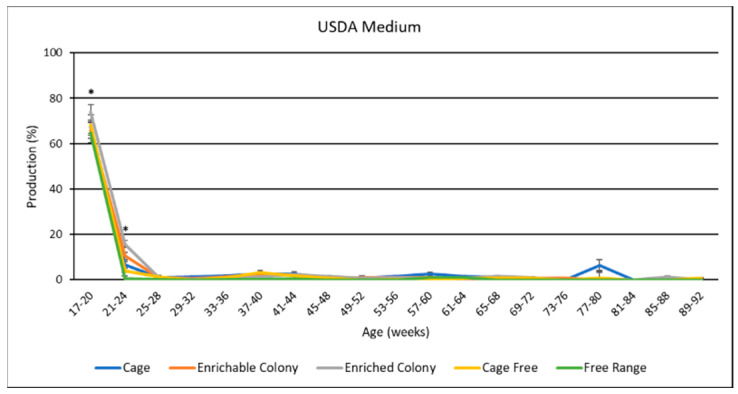
The effect of housing system on USDA medium production of brown egg layers by age. * Signifies a significant effect (*p* < 0.05) of housing environment during that age range on USDA medium percentage.

**Table 1 animals-13-00694-t001:** Housing system and replicate allocation of strains during the lay cycle.

Strain	Caged	Enrichable	Enriched	Cage-Free	Free-Range
Hy-Line Brown	8 (4)	6 (3)	6 (3)	4 (2)	4 (2)
Hy-Line Silver Brown	8 (4)	6 (3)	6 (3)	2 (2)	2 (2)
Lohmann LB-Lite	8 (4)	6 (3)	6 (3)	4 (2)	4 (2)

Numbers indicated within ( ) are post molt replications (after period 14).

**Table 2 animals-13-00694-t002:** Feeding program of diets according to egg production rate and *ad libitum* consumption rate.

Rate of Production	Feed Consumption g/100 Birds/Day	Diet Fed
Pre-production	<9.52	Pre-Lay
	<10.43	Pre-Peak
Pre-peak and >90%	10.43–12.20	Layer 1
	>12.20	Layer2
	<11.29	Layer 2
90–80%	11.29–12.20	Layer 3
	>12.20	Layer 4
	<11.29	Layer 4
70–80%	11.29–12.20	Layer 5
	>12.20	Layer 6
	<11.29	Layer 6
<70%	11.29–12.20	Layer 7 ^1^
	>12.20	Layer 7 ^1^

^1^ Layer 7 did not get used during this study.

**Table 3 animals-13-00694-t003:** Ingredient composition and calculated nutrient analysis of diets fed to all hens according to the feeding program described in Table 2.

Ingredients	Pre-Lay	Pre-Peak	Layer 1	Layer 2	Layer 3	Layer 4	Layer 5	Layer 6
	(%)	(%)	(%)	(%)	(%)	(%)	(%)	(%)
Corn	48.7	58.3	60.1	62.0	68.0	66.5	65.8	65.2
Soybean Meal	35.2	28.2	26.7	25.3	25.0	22.0	20.9	18.9
Wheat Midds	-	-	-	-	-	-	5.70	12.9
Fat (Lard)	0.55	0.50	-	-	0.83	-	-	-
Soybean Oil	2.54	1.29	1.81	1.25	0.095	-	-	-
Lysine 78.8%	-	-	-	-	-	0.11	0.005	-
D.L. Methionine	0.17	0.15	0.12	0.10	0.095	0.078	0.062	0.057
Ground Limestone	6.87	6.12	6.08	5.53	-	5.78	5.96	6.18
Course Limestone	3.87	3.50	3.5	3.75	3.97	3.75	3.75	3.75
Bi-Carbonate	0.11	0.10	0.10	0.15	0.11	0.10	0.10	0.10
Phosphate mono/D	1.21	1.07	0.90	1.30	1.26	1.09	0.99	0.82
Salt	0.39	0.32	0.29	0.25	0.31	0.26	0.26	0.24
Vit. Premix ^1^	0.05	0.05	0.05	0.05	0.05	0.05	0.05	0.05
Min. Premix ^2^	0.05	0.05	0.05	0.05	0.05	0.05	0.05	0.05
HyD3 Broiler (62.5 mg/lb)	-	-	0.025	-	-	-	-	-
Prop Acid 50% Dry	0.055	0.05	0.05	0.05	0.053	0.05	0.05	0.05
T-Premix	0.055	0.05	0.05	0.05	0.053	0.05	0.05	0.05
0.06% Selenium Premix ^3^	0.055	0.05	0.05	0.05	0.053	0.05	0.05	0.05
Choline Cl 60%	0.090	0.097	0.080	0.050	0.046	0.026	0.005	-
Avizyme	0.055	0.050	-	-	-	-	-	-
Ronozyme P-CT 540%	0.022	0.020	0.020	-	-	-	-	-
Calculated Values								
Crude Protein %	19.43	18.1	17.5	17	16.37	15.87	15.49	14.93
Calcium %	4.1	4.05	4	3.95	3.95	4	4.05	4.1
A. Phos. %	0.45	0.44	0.4	0.38	0.35	0.33	0.31	0.28
Total Lysine %	1.1	1	0.96	0.91	0.87	0.91	0.8	0.75
Total Sulfur Amino Acids %	0.8	0.74	0.69	0.66	0.63	0.6	0.58	0.56
ME kcal/kg	2926	2904	2860	2843	2843	2822	2800	2778

^1^ Vitamin premix supplied the following per kilogram of feed: vitamin A, 26,400 IU; cholecalciferol, 8000 IU; niacin, 220 mg; pantothenic acid, 44 mg; riboflavin, 26.4 mg; pyridoxine, 15.8 mg; menadione, 8 mg; folic acid, 4.4 mg; thiamin, 8 mg; biotin, 0.506 mg; vitamin B12, 0.08 mg; and ethoxyquin, 200 mg. The vitamin E premix provided the necessary amount of vitamin E as DL-α-tocopheryl acetate. ^2^ Mineral premix supplied the following per kilogram of feed: 120 mg of Zn as ZnSO_4_·H_2_O, 120 mg of Mn as MnSO_4_·H_2_O, 80 mg of Fe as FeSO_4_·H_2_O, 10 mg of Cu as CuSO_4_, 2.5 mg of I as Ca(IO_3_)_2_, and 1.0 mg of Co as Co_S_O_4_. ^3^ Selenium premix provided 0.3 ppm Se from sodium selenite.

**Table 4 animals-13-00694-t004:** Effect of housing system environment on production parameters of brown egg layers.

Housing Environment	Body Weight (kg/Bird)	Hen-Day Prod. (%)	Hen-Housed Prod. (%)	Feed Cons. (g/Bird/Day)	Feed Conv. (Egg g/Feed Cons)	Egg Weight (g)	Mortality (%)
Caged	2.07 ^B^ ± 0.03	78.7 ^B^ ± 0.3	72.9 ^C^ ± 0.4	105.7 ^B^ ± 0.4	0.452 ^B^ ± 0.002	60.5 ^C^ ± 0.1	19.1 ^AB^ ± 5.9
Enrichable Colony	2.05 ^B^ ± 0.03	81.2 ^A^ ± 0.3	71.9 ^C^ ± 0.5	113.0 ^A^ ± 0.4	0.443 ^BC^ ± 0.003	61.1 ^B^ ± 0.1	36.9 ^A^ ± 6.8
Enriched Colony	2.06 ^B^ ± 0.03	80.8 ^A^ ± 0.3	76.6 ^B^ ± 0.5	112.1 ^A^ ± 0.4	0.442 ^C^ ± 0.003	61.1 ^B^ ± 0.1	19.4 ^AB^ ± 6.8
Cage-free	2.1 ^AB^ ± 0.04	79.1 ^B^ ± 0.4	72.2 ^C^ ± 0.7	105.4 ^B^ ± 0.6	0.465 ^A^ ± 0.003	61.0 ^BC^ ± 0.2	17.2 ^AB^ ± 8.3
Free-range	2.21 ^A^ ± 0.04	82.0 ^A^ ± 0.4	80.0 ^A^ ± 0.7	113.1 ^A^ ± 0.6	0.467 ^A^ ± 0.003	63.7 ^A^ ± 0.2	6.7 ^B^ ± 8.3
*p*-Value	0.0206	0.0001	0.0001	0.0001	0.0001	0.0001	0.0039

^A,B,C^ Mean values within a column with different letter superscripts are significantly different (*p* < 0.05).

**Table 5 animals-13-00694-t005:** The effect of housing system environment on USDA grades and sizes of brown layers.

Environment	USDA Grades			USDA Egg Sizes			
	A%	B%	Loss%	XL%	L%	M%	S%
Caged	90.2 ^B^ ± 0.4	0.78 ^B^ ± 0.1	8.99 ^B^ ± 0.4	60.4 ^C^ ± 0.6	33.1 ^A^ ± 0.6	5.12 ^A^ ± 0.2	1.44 ± 0.2
Enrichable	88.6 ^C^ ± 0.4	0.92 ^AB^ ± 0.1	10.48 ^A^ ± 0.4	65.2 ^B^ ± 0.7	28.5 ^B^ ± 0.6	4.67 ^AB^ ± 0.3	1.65 ± 0.2
Enriched	87.8 ^C^ ± 0.4	1.16 ^AB^ ± 0.1	11.00 ^A^ ± 0.4	64.0 ^B^ ± 0.7	28.5 ^B^ ± 0.6	5.48 ^A^ ± 0.3	1.23 ± 0.2
Cage-free	96.6 ^A^ ± 0.6	1.39 ^A^ ± 0.2	2.06 ^C^ ± 0.5	63.8 ^B^ ± 0.9	30.2 ^B^ ± 0.8	4.50 ^AB^ ± 0.4	1.52 ± 0.2
Free-Range	97.8 ^A^ ± 0.6	0.66 ^B^ ± 0.2	1.53 ^C^ ± 0.5	77.0 ^A^ ± 0.9	18.1 ^C^ ± 0.8	3.55 ^B^ ± 0.4	1.40 ± 0.2
*p*-Value	0.0001	0.0172	0.0001	0.0001	0.0001	0.0005	0.3925

^A,B,C^ Mean values within a column with different letter superscripts are significantly different (*p* < 0.05).

## Data Availability

Data available upon request.

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
