# Peer review of "The Effect of Housing Environment on Commercial Brown Egg Layer Production, USDA Grade and USDA Size Distribution"

_animals, 2023, doi:10.3390/ani13040694_

Round 1

Reviewer 1 Report

The authors performed a comprehensive comparative study to investigate the correlation between housing environments and brown egg layer production. Due to the increasing interest in the well-being of egg layers and the adoption of various housing systems in the industry, a clear understanding of the effect of the housing environment on numerous parameters of egg layer production is a timely and important research topic. The study design is sound and numerous egg production parameters were thoroughly investigated. All sections were clearly presented with close attention to important details. This manuscript is almost ready to publish in my opinion except for only a few minor corrections that require some attention from the authors.

1. When tables or figures were cited in the main text, sometimes the first letter is a capital letter, while most times in a small letter. I think it is O.K. either way, but consistency is important. I suggest checking all figures and tables cited in the text and to make appropriate changes.

2. Line 115: change "rep" to "Housing system".

3. Line 128: change "were" to "was".

4. In Table 2 legend, italicize "ad libitum".

5. On page 6, there are multiple formulas are embedded into the text. Please check the indentation and make changes to make them consistent.

6. Line 412. I suggest changing "be" to "have".

7. Line 587. Remove "be" so that it would read "..the house can benefit...".

8. There are 5 housing systems. In most places, acronyms were used. I found in some places the full names were used instead with no clear reasons. Please check and make them consistent throughout.

Very nice work!

Author Response

Hello Reviewer 1,

Thank you for giving my manuscript a read and for your comments. They have helped me improve the manuscript.  I hope you find the edits to your satisfaction.

I have copied and pasted your comments and my responses below them.

Thanks for your help,

Ben

  1. When tables or figures were cited in the main text, sometimes the first letter is a capital letter, while most times in a small letter. I think it is O.K. either way, but consistency is important. I suggest checking all figures and tables cited in the text and to make appropriate changes.

Thank you for pointing this out. I have gone with lower case for tables.

  1. Line 115: change "rep" to "Housing system".

Changed rep to housing system.

  1. Line 128: change "were" to "was".

Changed were to was

  1. In Table 2 legend, italicize "ad libitum".

Italicized ad libitum

  1. On page 6, there are multiple formulas are embedded into the text. Please check the indentation and make changes to make them consistent.

Thank you for pointing this out. I am unsure what was happening there. I reapplied the equations into the equation format from the template. Looks like the format has been fixed now.

  1. Line 412. I suggest changing "be" to "have".

Changed be to have

Reviewer 2 Report

1. In the Introduction chapter, the hypothesis posed in the undertaken research should be clearly presented. Please do not refer to quotations from the literature in your hypothesis. 2. Materials and methods – were the hens to be tested from the same farm? What were they fed until 17 weeks of age? Could this have affected the subsequent laying of hens? 3. nutritional analysis in table 2 is required - the lack of this data does not allow to assess whether the feed mixtures were actually similar or the same in all systems of laying hens 4. Discussion - Text can be improved to be more concise 5. Conclusion - should be reworded and clearer

Author Response

Hello Reviewer 2,

Thank you for your comments and suggestions. Below are your comments and my responses to them. I appreciate your assistance in bettering my manuscript. I hope you will find my edits acceptable.

Benjamin Alig

  1. In the Introduction chapter, the hypothesis posed in the undertaken research should be clearly presented. Please do not refer to quotations from the literature in your hypothesis.

Thank you for pointing this out. I have rewritten the hypothesis so that is does not include citations and I have also attempted to make it clearer. I have also added a sentence near the beginning of that paragraph about those papers that I had cited.

  1. Materials and methods – were the hens to be tested from the same farm? What were they fed until 17 weeks of age? Could this have affected the subsequent laying of hens?

Thank you for addressing this. I have added a sentence that all houses were part of the same location and that all hens were fed the same feed during the pullet and rearing phase. I did not go into detail about the pullet phase as this is not what the study was focused on however, I also added a citation to a report on the pullet phase if readers are interested in learning more about how the hens were raised.

  1. nutritional analysis in table 2 is required - the lack of this data does not allow to assess whether the feed mixtures were actually similar or the same in all systems of laying hens

I did not include a nutritional analysis as this is not a nutritional study. Furthermore, since all environments were part of the same farm, all hens were fed the same feed and from the same batch of feed. Therefore, even if the analyzed values were different from the calculated values then ,all of the hens would be consuming the same feed.

  1. Discussion - Text can be improved to be more concise

I attempted to remove some repetitive sentences and reword some other sentences in the discussion. I realize that I restated much of the results and some of it I did not address in the discussion. I have attempted to simplify the reiteration of the results in the discussion. I have also broken up the discussion section into subsections like the results section for ease of reading.

  1. Conclusion - should be reworded and clearer

Thank you for the suggestion to retool the conclusion. I rewrote the conclusion to make it clearer. I hope the rewrite easier to read.

Round 2

Reviewer 2 Report

none